# What Does the Feeling of Job Success Depend On? Influence of Personal and Organizational Factors

Susana Rubio-Valdehita * , Eva María Díaz-Ramiro and María Inmaculada López-Núñez

Faculty of Psychology, Universidad Complutense de Madrid, 28223 Madrid, Spain; ediazram@ucm.es (E.M.D.-R.); mariai04@ucm.es (M.I.L.-N.)
* Correspondence: srubiova@ucm.es

**Abstract:** The main objective of this study was to determine the influence that various personal and organizational factors have on the self-assessed performance of 1459 employees recruited through a convenience sampling technique. The self-assessed performance was used as indicator of the feelings of job success. A non-experimental correlational cross-sectional design was established. Measures of the sociodemographic characteristics of the participants (such as age, gender, marital status, and personality), structural features of the organization (such as national vs. international, number of employees, or professional sector), and psychosocial aspect of the jobs (such as workload or burnout) were collected via a Google Form Questionnaire. Data were explored using multiple stepwise regression. Results showed conscientiousness as the most important predictor of perceived job success, followed by performance demands and personal accomplishment. Age, extraversion, and having a permanent contract were also related to better perceived success. The main conclusion is that perceived success is greater in the conscientious, extroverted, older participants, with a stable employment contract who have a job with high responsibility, and that provides them with greater feelings of personal fulfillment. The practical implications as well as the strength and limitations of the study are described.

**Keywords:** job success; personality; age; gender; psychosocial factors; workload; conscientiousness; self-assessed performance

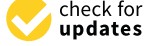



## 1. Introduction

At work, all people apply their skills and competencies hoping that success will come as a consequence of a good performance, and therefore of a satisfactory result in carrying out their work tasks [1]. Two complementary forms of professional success have traditionally been distinguished: objective and subjective/perceived success [2]. The first refers to quantifiable and measurable aspects such as salary, status, professional category, and professional career, while the second are related to more qualitative aspects depending on individuals' assessments of their professional experience, and on the comparisons that the individual makes with the rest of the employees.

Studies focused on evaluating objective success have found it difficult to consider aspects such as status or salary in a non-biased way, which are not a priority in certain professions, differences between men and women, or intercultural variations regarding the organic structures of companies and employee development plans [3]. Some research [4] suggests that there is an interdependence between objective and subjective success, indicating that objective success does not always lead to subjective or perceived success, but rather that perceived success precedes objective success. Subjective job success reflects an individual's internal evaluation of his or her career, across every element that is perceived as important by the individual [2], and it has become particularly important in the current work environment because only individuals themselves can meaningfully define and assess their success with reference to their own self-defined standards, needs, values and

aspirations. Subjective professional success is important as it has consequences on several beneficial organizational outcomes, such as productivity, employee commitment, or organizational retention [5,6]. Here, we present a study carried out to verify the possible influence of several personal and organizational factors on the self-assessed job performance of a sample of employees in different professional sectors.

## 2. Related literature Review

The interest in studying which factors influence perceived success at work is long-standing [7,8]. However, research in the field has not yet achieved clear results. Several models have been proposed, including employee factors (such as attitudes, gender, age, professional experience, education, level of commitment, personality profile, or professional development), and organizational context variables (such as social support, job context, job demands, or company structure) [8].

### *2.1. Personal Factors*

#### 2.1.1. Personality

One personal factor that has received the most attention in job success research is personality, especially following the Big Five model [9–12]. The Big Five taxonomy of personality proposes five factors to encompass personality: extraversion, neuroticism, conscientiousness, openness to experience, and agreeableness. Extraverted people are optimistic, sociable, and adventurous. Neurotics are emotionally unstable, anxious, and insecure. Conscientiousness is related to trustworthy, detailed, meticulous, and organized people. Openness to experience is shown by innovative, non-conformist, flexible, and open-minded people. Agreeableness is typical of people who are friendly, trustworthy, helpful, and cooperative. Conscientiousness, openness to experience, extroversion, and emotional stability tend to be positively associated with both objective (salary, promotion) and subjective (job satisfaction) measures of success [13]. A study [14] summarizing the results of 15 prior meta-analytic studies about the relationship between the Five Factor Model personality traits and job performance supports that conscientiousness is a great predictor across performance measures in all occupations studied. Neuroticism was also found to be a generalizable predictor when overall work performance was the criterion, but its relationship to specific performance measures and occupations was less consistent than conscientiousness. However, extraversion, openness, and agreeableness only predict success in certain occupations or relate to specific criteria.

Hurtz and Donovan [9] provided a meta-analytic estimate of the criterion-related validity of Big Five measures for predicting task performance and contextual performance (work commitment and interpersonal facilitation). Their results showed that conscientiousness and emotional stability predicted all three performance criteria, whereas agreeableness was more solely related to interpersonal facilitation, and they concluded that although agreeableness does not influence task performance, it does appear to influence interpersonal facilitation. It should be noted, however, that none of these analyses for the task and contextual performance criteria revealed stronger actual validations than the overall performance analysis.

Kamdar and Van Dyne [11] examined the effects of employee personality, specifically conscientiousness and agreeableness, in predicting the performance of 230 engineers and found that these two personality factors are desirable and directly related to performance, but especially when social exchange relationships with supervisors and co-workers are not of high quality.

#### 2.1.2. Gender

Studies on the influence of gender on perceived job success have focused mainly on three aspects: one is related to the impact of family duties (maternity, child care, and prioritization of the husband's career) [15], which leads women to opt for part-time jobs and relocation to a greater extent than men; the second reason could be that women show lower

managerial ambitions compared to men, due to the social roles expected of women [16]; and the third problem is related to gender discrimination in the workplace, whereby women are prevented from achieving successful careers because they receive differential treatment from organizations [17].

Some authors [18] propose that gender has a clear incidence on work success, while other [19] do not find significant differences between men and women. Arenas et al. [20] analyzed the effect of gender on performance in a complex decision-making task, comparing self-regulation processes and dispositional factors between male and female university students when performing a task that simulates the implementation of innovation and change in an organization. Their results show that women show a clear tendency to communicate mistakes and are less concerned with demonstrating competence to others. However, when faced with difficulties, women feel less capable of dealing with difficulties, feeling more pained and less confident in their performance, which leads them to achieve lower perceived success than men [20]. Moreover, Price-Glynn and Rakovski [21] points out that women are more likely to have lower performance in competitive situations [22]. It has been confirmed [23] that gender stereotypes significantly influence the perceived performance of women, and that gender modulates the relationship between structural variables (especially mobility and dedication) and objective success, while psychosocial variables (such as workload) determine perceived success, job satisfaction, and psychological well-being [24].

### 2.1.3. Age

Research on the influence of age on perceived success is insufficient and has yielded controversial results. While some have found positive relationships between age and success, others have found a negative relationship, and others conclude that age has no significant effects [25]. This may be because the relationship between age and job performance is mediated by opportunities, individual goals, options, and future possibilities [26]. For example, Andrade and Westover [25] concluded that age has a statistically significant positive impact on perceived job success (the older you get, the more satisfied you are with your job), so there are statistically significant intergenerational differences in perceived success levels across generations. However, Alessandri et al. [26] carried out a longitudinal study on the evolution of work performance using a sample of 420 employees followed up over a period of six years, finding a non-linear trajectory with alternating phases of stability, rapid increase, and abrupt decrease. It was found that job tenure, level of education, perception of the direct supervisor, and self-efficacy were significant predictors of job performance [26].

In a recent meta-analysis on the influence of age on entrepreneurial success [27], it was shown that age had a weak positive linear relationship with overall success, but that the relationship is negative among the youngest and positive between the oldest participants. The size of the positive effect increases when there are more women in the sample. The effect of age was similar independent of the seniority of the participants. Regarding the type of measures of success, age had a negative effect on objective performance but a positive effect on subjective success. When a negative relationship between success and age has been found, explanations have been based on the poorer general health of the older participants, greater psychological rigidity and resistance to change, less risk-taking, and a pessimistic social perception towards older people. On the other hand, the perspectives that give reasons for a positive relationship between age and professional success are based on the accumulation of knowledge and wisdom of the elderly, and on the greater emotional regulation that comes from their life and work experience. Nevertheless, it is not clear whether these advantages of older workers can compensate for their weaknesses and the social prejudices against the elderly. Several studies [26,28] have found that older workers are perceived positively in terms of crystallized intelligence and conscientiousness, and that they show greater commitment to organizations, while the younger workers are viewed positively in terms of their fluid intelligence and proactive personality. Other research [29] has indicated that older employees show higher professional performance, but are more



resistant to change, which makes it difficult for them to adapt in situations of innovation or organizational change. However, older workers trained by their organization and with more work experience would show better job performance [29].

### 2.1.4. Other Personal Factors

Other personal factor, that to some extent are related to those previously mentioned, are marital status, having children, education, or reason for taking the job. In general, research have found that males who are married and have a higher education level have more perceived success [13,30]. A positive relationship between education level and job satisfaction has been found [31], however, tenure was inversely related to career satisfaction [32]. On the other hand, married employees in general [13], and married women in particular [33], are more satisfied than those who are not married.

### *2.2. Work Factors*

The human resources management (HRM) in organizations takes an important interest in evaluating the performance of workers as an indicator of the success of the company itself [34]. Research on the management and development of people in organizations [35–38] suggests that HMR practices influence performance by affecting the abilities, motivation, and opportunities of employees to use their skills. However, understanding of the factors and processes that potentially mediate the relationship between HRM and performance remains limited [39–43]. Even though more than 91% of companies include performance management plans among their HRM strategies [44], both managers and employees are dissatisfied with their performance management processes [45,46]. Only 30% of employees report that their organization's performance management system supports them to improve their performance [47], so it is necessary to determine which specific variables allow performance to be predicted. Two groups of variables have been proposed as the main work factors linked to professional success: structural and psychosocial [24].

### 2.2.1. Structural Factors

Factors such as organizational size also affect career success [48], as organization size positively relates to number of promotions. It is thought that larger organizations have a greater ability to pay and offer more promotion opportunities, leading to a higher perception of success [49]. However, Horwitz et al. [50] found no significant difference based on total number of employees, but financial factors such as sales turnover were better indicators of choosing a particular human resources strategy than workforce size. The determinants of professional success come from organizational and individual factors. Organizational factors include employee perception of development opportunities provided by the organization, and clarity of the roles. Staying in the same position for a long time generates a feeling of stagnation and has a negative influence on professional satisfaction. While different measures of success have different determinants, it is clear that career success depends on the actions of both the organization and the individual [49].

Significant differences in perceived success have been found between the nature of the business, concluding that workers in the transportation and storage sector, salespeople, and clerks showed a higher risk of lower success [51]. However, other studies have not found differences in the professional success perceived by workers from different professional sectors [9,11].

The possible effect on job success of temporary employment that comes from the type of employment contract (temporary or permanent) that the organization makes to its employees is also not clear. There are very few systematic studies on the effects of temporary contracts, and the results of those studies are inconsistent. While some studies support the view that temporary contracts negatively affect employee outcomes, such as mental health and satisfaction, other studies show that a temporary contract status may be associated with lower job strain [52]. The results suggest that a temporary employment condition is associated with positive and negative consequences. On the negative side,

temporary status reduced perceptions of job security and participatory decision-making, which has detrimental effects on job strain. On the other hand, temporary workers have less of a workload [53].

### 2.2.2. Psychosocial Factors

Several studies based on the Demands and Resources model [54] have studied the weight of job demands and resources in predicting variables such as absenteeism [55] and work performance [56]. Job demands refer to those physical, psychological, social, or organizational aspects of work that require a sustained physical and/or psychological (cognitive and emotional) effort and that, therefore, are associated with certain physiological and/or psychological costs for the employee. Some examples are high work pressure with overload of functions, high emotional demands, and inadequate environmental conditions. Job demands have an evident influence on job success, since situations of inadequate workload, either due to excess or lack of demands, can have important negative consequences for performance and health [57]. Other studies have explored how perceived performance depends on the cognitive demands and the perceived quality of the work context [58]. The results have shown that an increase in the worker's cognitive demand translates into a direct effect on an increase in performance. Regarding the quality of work context, the results indicate that the increase in the perceived quality of working life experienced by the worker translates into a direct effect on the improvement of the employee performance. Despite these studies, few have analyzed the effects of job demands on the perception of performance expressed by workers, specifically as an indicator of subjective success. Most of the studies analyze the relationship between psychosocial factors and worker's health and consider the psychological health of workers as a measure of professional success and not as a variable that makes it possible to predict such success [59]. However, it has been found that excess of psychological demands at work, small job autonomy, as well as lack of social support, are predictors of declines in job success [60]. Job insecurity, lack of organizational support, job dissatisfaction, and worker' health problems have also been found to predict subjective discomfort and feelings of unsuccess [61]. Six key areas in which these imbalances take place have been identified: workload, control, reward, community, fairness, and values. Mismatches in these areas affect an individual's level of experienced burnout, which in turn determines various outcomes, such as job success, social behaviors, and personal wellbeing [60].

Understanding the way in which individual and organizational factors influence the perceived performance of workers is a key element for the design of HRM strategies, but in the reviewed literature there is controversy on which variables can predict perceived professional success. Furthermore, most of the studies have considered professional success as an objective indicator and not as a subjective value. Therefore, the main goal of this research was to examine subjective performance and determine which specific variables make it possible to predict perceived job success based on two broad categories: personal and organizational factors. In our study, we analyzed the predictive value of multiple personal, structural, and organizational variables with respect to perceived job success.

## 3. Materials and Methods

### 3.1. Participants

A sample of 1459 participants was used. All of them were employees of different professions in Spain. Table 1 shows the distribution of the participants based on sociodemographic and work characteristics. The age of the participants ranged from 18 to 67 years (Mean= 34.00, *SD* = 11.08).

**Table 1.** Characteristics of the sample and mean and standard deviation (*SD*) of success.

|  |  | *N* | **Mean** | *SD* |
|---|---|---|---|---|
| Sex | Woman | 933 | 8.29 | 1.38 |
|  | Man | 526 | 8.41 | 1.20 |
| Education | Primary | 93 | 8.61 | 1.18 |
|  | Secondary | 283 | 8.36 | 1.47 |
|  | Higher | 1083 | 8.31 | 1.29 |
| Marital status | Single | 750 | 8.22 | 1.36 |
|  | Married/in couple | 635 | 8.46 | 1.22 |
|  | Separated/divorced | 74 | 8.38 | 1.62 |
| Children | No | 1033 | 8.25 | 1.33 |
|  | Yes | 426 | 8.54 | 1.27 |
| Seniority | <1 years | 413 | 8.02 | 1.36 |
|  | 1–2 years | 233 | 8.44 | 1.23 |
|  | 2–5 years | 331 | 8.38 | 1.19 |
|  | >5 years | 482 | 8.48 | 1.37 |
| Motivation | Necessity | 463 | 8.22 | 1.38 |
|  | Vocational | 996 | 8.39 | 1.28 |
| Contract | Temporal | 506 | 8.07 | 1.40 |
|  | Permanent | 953 | 8.48 | 1.25 |
| Work on weekends | No | 658 | 8.22 | 1.39 |
|  | Yes | 801 | 8.43 | 1.25 |
| Sick leave | No | 1075 | 8.37 | 1.28 |
|  | Yes | 384 | 8.25 | 1.41 |
| Professional sector | Trade | 782 | 8.42 | 1.27 |
|  | Education | 240 | 8.35 | 1.27 |
|  | Administration and finance | 124 | 8.10 | 1.37 |
|  | Health | 161 | 8.08 | 1.57 |
|  | Industry | 152 | 8.32 | 1.25 |
| Size of the company | Small | 406 | 8.28 | 1.37 |
|  | Median | 427 | 8.27 | 1.33 |
|  | Big | 626 | 8.24 | 1.35 |
| Business field | National | 1045 | 8.31 | 1.35 |
|  | International | 414 | 8.14 | 1.33 |

*3.2. Design and Procedure*

A non-experimental correlational cross-sectional design was established. The target population was the Spanish working population. We recruited the sample of participants through LinkedIn, Instagram, and Twitter using the convenience sampling technique, since participation was completely anonymous and voluntary. Thus, participants accessed a link that led to a Google Forms questionnaire. All participants gave their informed consent to participate. The ethics committee of the authors' research center approved this study (Ref.: 2019/20-022).

*3.3. Measures and Instruments*

3.3.1. Personal Factors

A sociodemographic data questionnaire was used, in which participants were asked about their gender (0: female and 1: male), age, marital status (1: single, 2: married or in a relationship, and 3: separated/divorced), if they had children or not, and the reason for which they took up their job (1: because they like it, for personal or professional development and 0: due to economic needs or difficulty finding another job). The NEO PI-R (adapted for the Spanish population) was applied to evaluate personality [62]. This questionnaire assesses the Big Five personality factors: neuroticism, extroversion, openness to experience, agreeableness, and conscientiousness.

3.3.2. Organizational Factors

a.　Structural factors: participants were asked about the type of contract (0: temporary and 1: permanent), seniority in the position, whether their working hours included working on weekends, the size of the company, the professional sector, and the field of activity (national vs. international).

b.　Psychosocial factors: several job psychosocial factors were assessed: burnout, workload, social support, job autonomy, and reward satisfaction. The MBI-GH (Spanish version) [63] was applied to evaluate burnout in a three-dimensional model (emotional exhaustion, depersonalization, and personal accomplishment). MBI-GH is the most common and suitable instrument to assess burnout. Using this 22-item tool, responders rate the frequency with which they experience various feelings or emotions on a 7-point Likert scale, with response options ranging from "Never" to "Daily". Higher values of depersonalization (DP) and emotional exhaustion (EE) and lower values of personal accomplishment (PA) signify burnout. Workload was evaluated by applying the CarMen-Q Questionnaire [64]. CarMen-Q assesses the demands of the job in a four-dimensional model, including aspects related to task demands (cognitive, temporal, and performance demands) and subject experience (emotional demands). CarMen-Q has 29 items with a 4-point Likert scale to rate the frequency with which employees experience their working conditions, ranging from "Never" to "Always". The cognitive demands dimension refers to attentional, complex information processing, and decision-making aspects required by the job. The temporal demands dimension includes aspects related to the pace of work and speed demands. The performance demands dimension takes account for performance requirements and the job's degree of responsibility. The emotional demand dimension of the CarMen-Q includes aspects such as the degree to which the job makes the worker nervous, anxious, or stressed. In addition, questions about job autonomy, support, and reward satisfaction were included. Seven items (on a 5-point Likert scale from "Never" to "Always") were used for each of these variables. Items used to assess job autonomy were: "I have the freedom to decide how to do the work" or "I can decide my work schedule with flexibility" (Cronbach' $\alpha$ = 0.83). For support: "Relationships with my peers are good" or "My bosses help me if I have problems with the job" (Cronbach' $\alpha$ = 0.78). For reward satisfaction: "I think the money I receive for doing my job is adequate " or "Prospects for future salary increases are good" (Cronbach' $\alpha$ = 0.85).

3.3.3. Perceived Job Success

Perceived success was evaluated through a single-item scale from 0 to 10 in which participants were asked "How satisfied are you with the work performance you have achieved in the last year?".

*3.4. Data Analysis*

First, descriptive analyses of the variables and the relationships between them were performed. Since high relationships were found between some of the variables, a stepwise hierarchical regression analysis was subsequently carried out in which the feeling of success at work was the dependent variable and personal variables were introduced as independent variables in the first step, in the second step the structural aspects, and in the third step the psychosocial variables. The stepwise procedure was chosen to find the model that best fits the data, avoiding collinearity problems. Finally, the sample was divided into two groups based on their perceived level of success (high: success $\geq 9$; normal: success $\leq 8$), and a binary logistic regression analysis was performed in which the predictor variables resulting from the previous analysis were introduced as predictors. All analyses were performed with SPSS 27.0.

## 4. Results

Table 1 shows the mean and standard deviation of success among the qualitative variables of the study.

Table 2 shows the Pearson's correlation coefficients between the continuous variables of the study.

**Table 2.** Pearson's coefficients between the continuous variables of the study.

| | 1 | 2 | 3 | 4 | 5 | 6 | 7 | 8 | 9 | 10 | 11 | 12 | 13 | 14 | 15 | 16 | 17 |
|---|---|---|---|---|---|---|---|---|---|---|---|---|---|---|---|---|---|
| 1. Sucess | 1 | | | | | | | | | | | | | | | | |
| 2. Age | 0.15 ** | | | | | | | | | | | | | | | | |
| 3. Neuroticism | −0.20 ** | −0.19 ** | | | | | | | | | | | | | | | |
| 4. Extraversion | 0.20 ** | −0.03 | −0.38 ** | | | | | | | | | | | | | | |
| 5. Openness | 0.01 | −0.12 ** | 0.06 * | 0.23 ** | | | | | | | | | | | | | |
| 6. Agreabiliness | 0.15 ** | 0.10 ** | −0.27 ** | 0.27 ** | 0.18 ** | | | | | | | | | | | | |
| 7. Conscientiousness | 0.30 ** | 0.12 ** | −0.45 ** | 0.27 ** | 0.03 | 0.28 ** | | | | | | | | | | | |
| 8. Seniority | 0.13 ** | 0.56 ** | −0.13 ** | −0.03 | −0.12 ** | 0.03 | 0.08 ** | | | | | | | | | | |
| 9. Support | 0.13 ** | −0.12 ** | −0.24 ** | 0.27 ** | 0.04 | 0.20 ** | 0.17 ** | −0.08 ** | | | | | | | | | |
| 10. Autonomy | 0.07 ** | 0.13 ** | −0.24 ** | 0.17 ** | −0.11 ** | 0.09 ** | 0.12 ** | −0.05 * | 0.45 ** | | | | | | | | |
| 11. Rewards | −0.07 ** | 0.02 | 0.17 ** | −0.10 ** | 0.07 ** | −0.09 ** | −0.06 * | −0.03 | 0.41 ** | −0.45 ** | | | | | | | |
| 12. Cognitive demands | 0.09 ** | 0.17 ** | −0.02 | 0.04 | 0.02 | 0.02 | 0.15 ** | −0.17 ** | 0.08 ** | −0.01 | −0.02 | | | | | | |
| 13. Emotional demands | −0.05 * | 0.05 * | 0.40 ** | −0.20 ** | 0.09 ** | −0.08 ** | −0.06 * | −0.18 ** | 0.40 ** | −0.41 ** | 0.30 ** | 0.39 ** | | | | | |
| 14. Temporal demands | 0.06 * | 0.02 | 0.12 ** | −0.07 ** | 0.08 ** | −0.02 | 0.07 ** | −0.11 ** | 0.31 ** | −0.54 ** | 0.25 ** | 0.42 ** | 0.60 ** | | | | |
| 15. Performance demands | 0.16 ** | 0.10 ** | −0.02 | 0.05 * | −0.02 | 0.07 ** | 0.19 ** | −0.11 ** | 0.10 ** | −0.14 ** | 0.02 | 0.67 ** | 0.33 ** | 0.41 ** | | | |
| 16. Emotional exhaustion | −0.13 ** | −0.04 | 0.44 ** | −0.26 ** | 0.08 ** | −0.17 ** | −0.17 ** | −0.09 ** | 0.45 ** | −0.45 ** | 0.36 ** | 0.21 ** | 0.78 ** | 0.52 ** | 0.18 ** | | |
| 17. Depersonalization | −0.12 ** | −0.10 ** | 0.35 ** | −0.21 ** | 0.01 | −0.28 ** | −0.23 ** | −0.01 | 0.35 ** | −0.28 ** | 0.17 ** | 0.09 ** | 0.40 ** | 0.27 ** | 0.10 ** | 0.53 ** | |
| 18. Personal accomplishment | 0.29 ** | 0.14 ** | −0.29 ** | 0.35 ** | 0.12 ** | 0.26 ** | 0.35 ** | −0.14 ** | −0.28 ** | 0.12 ** | −0.15 ** | 0.26 ** | −0.04 | 0.08 ** | 0.28 ** | −0.17 ** | −0.22 ** |

** $p < 0.001$; * $p < 0.05$.

To test relationships between personal and organizational variables, a chi-squared test was used. We found an association between sex and professional sector ($\chi^2 = 86.91$, $p < 0.001$), there being a higher percentage of men in Industry and more women in Education and Health. Size of the company were linked to the Business field ($\chi^2 = 155.00$, $p < 0.001$), showing that the bigger companies are international. Professional sector and size of the company were also related ($\chi^2 = 318.06$, $p < 0.001$) as Education and Health were the sectors in which a median size was the most frequent, while in the other sectors there were more large companies. The type of contract was also related to professional sector ($\chi^2 = 94.67$, $p < 0.001$), as permanent contracts predominate in all sectors except Health.

No significant relations were found between sex and marital status, or having children or job seniority ($p > 0.1$)

To test differences in psychosocial factors and personality due to sex, means comparisons were computed. Sex differences were found in EE ($T = 3.26$, $p = 0.001$), PA ($T = 3.65$, $p < 0.001$), autonomy ($T = 6.20$, $p < 0.001$), rewards satisfaction ($T = 5.32$, $p < 0.001$), neuroticism ($T = 3.99$, $p < 0.001$), extraversion ($T = 3.91$, $p < 0.001$), openness ($T = 2.08$, $p = 0.038$), agreeableness ($T = 2.24$, $p = 0.025$), cognitive demands ($T = 3.19$, $p = 0.001$), emotional demands ($T = 5.64$, $p < 0.001$), temporal demands ($T = 2.71$, $p = 0.007$), and per-

formance demands ($T = 2.13$, $p = 0.033$). Table 3 shows the means and standard deviations of these variables for men and women.

**Table 3.** Means and standard deviations (*SD*) of psychosocial factors and personality for women and men.

| | Woman | | Man | |
|---|---|---|---|---|
| | **Mean** | *SD* | **Mean** | *SD* |
| Neuroticism | 69.42 | 30.84 | 62.41 | 32.92 |
| Extraversion | 47.15 | 34.14 | 54.37 | 33.39 |
| Openness | 55.79 | 33.57 | 52.14 | 31.43 |
| Agreeableness | 43.84 | 31.46 | 40.04 | 30.44 |
| Conscientiousness | 40.88 | 33.24 | 42.28 | 32.06 |
| Emotional exhaustion | 59.80 | 30.15 | 54.95 | 25.48 |
| Depersonalization | 49.71 | 29.65 | 48.08 | 29.48 |
| Personal accomplishment | 50.12 | 26.67 | 44.72 | 27.75 |
| Support | 61.19 | 23.14 | 62.24 | 20.96 |
| Autonomy | 43.00 | 24.12 | 50.81 | 22.47 |
| Rewards | 46.71 | 21.77 | 40.33 | 22.27 |
| Cognitive demands | 56.82 | 23.56 | 60.62 | 20.80 |
| Emotional demands | 47.35 | 25.85 | 39.98 | 22.80 |
| Temporal demands | 53.97 | 23.27 | 50.75 | 20.84 |
| Performance demands | 68.46 | 22.00 | 70.72 | 17.90 |

To test differences in psychosocial factors due to the professional sector, means comparisons were computed. Significant differences ($p < 0.05$) were found in all factors except organizational support. Table 4 shows the means and standard deviations (*SD*) of psychosocial factors for each professional sector.

**Table 4.** Means and standard deviations (*SD*) of psychosocial factors for each professional sector.

| | Trade | | Education | | Administration and Finance | | Health | | Industry | |
|---|---|---|---|---|---|---|---|---|---|---|
| | **Mean** | *SD* | **Mean** | *SD* | **Mean** | *SD* | **Mean** | *SD* | **Mean** | *SD* |
| Emotional exhaustion | 59.80 | 28.99 | 56.93 | 27.03 | 50.81 | 28.52 | 63.47 | 28.04 | 51.05 | 27.96 |
| Depersonalization | 52.90 | 29.42 | 39.05 | 28.23 | 47.98 | 28.86 | 50.22 | 29.38 | 45.43 | 29.57 |
| Personal accomplishment | 46.82 | 26.90 | 56.97 | 25.67 | 40.02 | 26.83 | 56.57 | 25.27 | 39.05 | 27.20 |
| Support | 60.92 | 22.10 | 61.75 | 23.58 | 63.17 | 21.71 | 61.32 | 22.12 | 63.51 | 22.81 |
| Autonomy | 44.36 | 23.39 | 39.62 | 21.26 | 57.85 | 23.52 | 42.89 | 25.67 | 56.32 | 21.93 |
| Rewards | 44.98 | 22.70 | 46.60 | 19.02 | 41.17 | 22.42 | 48.61 | 21.70 | 36.26 | 22.21 |
| Cognitive demands | 51.41 | 22.68 | 67.21 | 18.01 | 62.98 | 18.99 | 71.99 | 20.29 | 60.35 | 21.62 |
| Emotional demands | 45.12 | 25.17 | 49.74 | 23.79 | 37.17 | 25.87 | 48.24 | 23.71 | 36.97 | 24.12 |
| Temporal demands | 53.54 | 22.04 | 56.98 | 21.36 | 45.74 | 21.03 | 55.87 | 25.68 | 45.02 | 20.98 |
| Performance demands | 65.35 | 21.06 | 71.14 | 17.12 | 71.83 | 18.31 | 83.85 | 19.06 | 68.99 | 19.53 |

Table 5 shows the results that were statistically significant in each step of the regression analysis. The final model ($R^2 = 20.18$) showed significant effects for age, resulting that the older the worker, the greater the perceived success; conscientiousness (the greater the conscientiousness, the greater the sensation of success); and extraversion (more extroverted, more perceived success). The type of contract also was significant so employees with a permanent contract (Mean = 8.47, *SD* = 1.27)) were more successful than temporary ones (Mean = 8.07, *SD* =1.40) and participants who worked on weekends showed a greater sense of success (Mean = 8.43, *SD* = 1.25), than those who only work from Monday to Friday (Mean = 8.21, *SD* = 1.42). Significant effects of emotional exhaustion, personal accomplishment, and performance demands were also found.

**Table 5.** Results of hierarchical multiple regression of personal, structural, and psychosocial factors on perceived success.

|  |  | β | t | p |
|---|---|---|---|---|
| **Step 1** | Conscientiousness | 0.25 | 9.86 | 0.000 |
|  | Extraversion | 0.13 | 5.17 | 0.000 |
|  | Age | 0.12 | 4.75 | 0.000 |
| **Step 2** | Conscientiousness | 0.25 | 9.64 | 0.000 |
|  | Extraversion | 0.13 | 5.07 | 0.000 |
|  | Age | 0.10 | 3.89 | 0.000 |
|  | Work on weekends | 0.09 | 3.62 | 0.000 |
|  | Contract | 0.09 | 3.36 | 0.001 |
| **Step 3** | Conscientiousness | 0.18 | 6.98 | 0.000 |
|  | Extraversion | 0.07 | 2.59 | 0.010 |
|  | Age | 0.08 | 2.99 | 0.003 |
|  | Work on weekends | 0.10 | 4.00 | 0.000 |
|  | Contract | 0.09 | 3.40 | 0.001 |
|  | Personal accomplishment | 0.14 | 5.01 | 0.000 |
|  | Performance demands | 0.09 | 3.39 | 0.001 |
|  | Emotional exhaustion | −0.08 | −3.19 | 0.001 |

Once the variables related to perceived success were determined, and to better understand which of them could differentiate between those with a clear perception of success from those who did not, a binary logistic regression analysis was performed starting from the categorisation of participants into two groups (the successful, with success scores equal to 9 and 10) and the unsuccessful (the rest)). The results (Table 6) showed that emotional exhaustion and working on weekends did not serve to differentiate both groups, however, the rest of the variables did, allowing 66.00% of the participants to be correctly classified.

**Table 6.** Results of logistic multiple regression of personal, structural, and psychosocial factors on perceived success (1: usual success vs. 2: high success).

|  | B | Wald | Sig. | Exp(B) |
|---|---|---|---|---|
| Age | 0.13 | 8.00 | 0.005 | 1.02 |
| Extraversion | 0.17 | 6.04 | 0.014 | 1.02 |
| Conscientiousness | 0.28 | 36.40 | 0.000 | 1.06 |
| Work on weekends | 0.04 | 3.59 | 0.058 | 0.78 |
| Contract | 0.11 | 5.20 | 0.022 | 0.72 |
| Personal accomplishment | 0.25 | 10.10 | 0.001 | 1.01 |
| Emotional exhaustion | −0.08 | 0.76 | 0.382 | 1.00 |
| Performance demands | 0.16 | 12.47 | 0.000 | 1.48 |

## 5. Discussion

Most studies show factors that affect objective measures of success, such as salary or number of promotions. While objective measures are important in assessing how far a person's career has progressed, subjective measures are just as important, as people have job expectations beyond financial compensation or promotion [65]. There seems to be no consistent results showing which variables influence subjective job success. The

purpose of this study was to examine what type of personal and organizational variables are associated with the perception of job success from a sample of people who currently work in different professional sectors. Self-assessed job performance was used as an indicator of the perceived degree of success. As predictors, sociodemographic and personality variables of the workers were measured, as well as organizational and job variables, combined with psychosocial measures (demands, burnout, support, autonomy, and satisfaction with rewards).

The main results showed that, of the personal variables, the most significant dimensions were conscientiousness, extraversion, and age. Although men showed greater perceived success than women, the gender variable did not reach sufficient predictive value in the analyses. A possible explanation for the fact that previous studies have found differences between men and women [18] could be that these studies have focused more on analyzing variables related to family responsibilities, professional sector, or difficulties to obtain a promotion in the position (need for mobility, competitiveness, etc.) [16] without considering aspects related to personality and psychosocial factors at work (such as burnout, organizational support, rewards, etc.). Our study indicates that when personality is incorporated, the differences between men and women in their perceived success are reduced considerably. Other studies have also found no differences between men and women in perceived job success. For example, Supangco [49] found, using a Philippine sample, that gender did not explain variation in total compensation, number of levels from the company president, and career satisfaction.

Our results confirm that conscientiousness is the dimension most associated with perceived success [14]. In agreement with previous research [66], conscientiousness and extraversion appeared as predictors of success, conscientiousness being the most important of both. In this sense, more conscientious people show a higher perception of success, which is even higher when the worker is extroverted. Thus, for example, Witt [66] have shown that, when extraversion is present together with conscientiousness, subjective performance is better, but if conscientiousness is low, performance deteriorates. In this sense, the results of previous studies [11] stand out, highlighting that organizations can improve job performance by selecting employees with a high level of consciousness, and that high-quality social exchange relationships at job can compensate for the lack of this characteristic in employees. This has important practical implications because in a situation where a manager has an employee who is unscrupulous or agreeable, developing a high-quality social relationship could be an option for improving job performance.

We have found a direct relationship between age and perceived success, such that older people perceive greater professional success [29]. Previous studies have explained this relation since older workers have accumulated more human and social capital, which they have acquired over time through their professional and life experiences [67], and also have better emotional regulation, which allows them to have a greater tolerance for stress in adverse conditions; thus, they are less likely to fail [68]. In addition, the life span theory explains how adult development involves loss, growth, and reorganization of psychological functioning through times periods, balancing for gains and losses, and that adult interests and needs change over time [69]. Through a structural equation analysis using a multi-source data set from 147 companies, the results of Kunze et al. [29] suggest that human resource policies that increase age heterogeneity in companies have positive effects on work climate and success at work. Our results agree with this positive effect that age diversity has on organizations.

Other specific individual factors such as education and marital status have been found to predict subjective success in other studies [15], indicating, for example, that married employees in general, and married women in particular, are more satisfied than those who are not. However, our results did not confirm these relationships from a multidimensional approach. For example, in the study carried out by Valcour and Ladge [15], in which relationships were found between marital status or the number of children with subjective success, the sample was made up exclusively of working women mothers, which limits

the generalization of the results to the general working population. More in line with our results are those obtained by Punnett et al. [33] with a sample of 1146 successful women from different countries of the American continent (Canada, Chile, Mexico, USA, Brazil, Argentina, and the Antilles), according to which there were no differences in performance satisfaction according to occupation or country, and most of the demographic variables investigated did not have a significant relationship with perceived success. Only age and being married showed any relationship with perceived success, although small, being more associated with higher self-efficacy and need for achievement scores and a greater internal locus of control, all of them being personality-related variables.

Among organizational factors, we found positive associations between perceived success and having long-lasting employment, and working on weekends and having higher performance demands. Considering that a short-term or temporary contract has been associated to lesser job demands and workload [53], and that performance demands refer to the degree to which the worker cannot make mistakes when accomplishing his/her responsibilities, since the consequences of his/her errors are serious, it seems clear that the participants with jobs that involve more responsibility and that require greater involvement in their work (even during the weekends) feel more satisfied with their performance and therefore have a greater perception of success. This result agrees with previous research on work engagement, indicating a positive relationship between work commitment and job success [70].

Greater success was mainly associated with more personal accomplishment and less emotional exhaustion, both dimensions of the so-called burnout syndrome. The other burnout factor considered in this study, depersonalization, showed a negative relationship with success, but not in a statistically significant way. In conclusion, the working conditions with the highest risk of the worker suffering from burnout problems are associated with feelings of job disappointment. This result would support research that considers workers' psychological health variables, such as burnout and related variables, as indicators of perceived success [32,59]. All organizations are required to ensure the physical and psychological health of their employees, so our results emphasize the importance of developing adequate strategies for HRM and for prevention of psychosocial risks at work. In contrast with previous research [60], although a positive correlation was found between social support and the perception of success at work, this was not statistically significant in the multivariate analysis and the relationship between autonomy and success was low.

## 6. Conclusions

This paper identifies some factors affecting job success using subjective measures. The main conclusion is that perceived success is greater in the conscientious, extroverted, older participants with stable employment who have a job with high responsibility, and that provides them with greater feelings of personal fulfillment.

Research on perceived job success is very important to both the individual and the organization. For individuals, job success is a logical expectation since they dedicate about a third of their time to work and have normally invested significant efforts in their professional training. For organizations, if employees have achieved their professional goals, this implies that the organization has also benefited from the performance of its workforce, which has given it a great competitive advantage.

To achieve professional success, both the individual and the organization spend time, effort, and resources in professional development actions. Although professional development is the joint responsibility of the individual and the organization, some organizational activities such as downsizing or reorganization, which result in fewer opportunities for employees, make professional development more difficult. In a constantly changing context, it is ever more essential for organizations to be healthier and for individuals to assume a more proactive role in the development of their feelings of success.

We believe that our study has as its main strengths that it is carried out with a large sample of workers from various professional sectors, with a multivariate analysis

perspective, since the importance of multiple personal and organizational factors is jointly analyzed and that, as far as we know, this is the only study of these characteristics that has been carried out with Spanish workers. However, it also has limitations. First, we used an incidental sample, which, although large, cannot be considered representative of the Spanish working population. In addition, this is a cross-sectional study, not longitudinal or experimental, so we can only draw conclusions about correlation relationships between variables and not about cause–effect relations. The final regression model only explained 20% of the variance of perceived success, which is small. A possible explanation may be that all the participants had a job, so very little variability was found in self-perceived success. It would be convenient in future studies to include participants who are unemployed or looking for a job in the sample, since this will probably increase the variability in the perception of job success. In addition, it would be very convenient to incorporate other variables that have not been taken into account here, such as work hours, the company's human resource management policies, the hierarchical level of the participant's job, or the influence of gender in traditionally masculinized or feminized professions [71,72].

Our study may have important practical implications for human resource management, since it shows the convenience of giving meaning and content to the job, with high performance demands that make the worker feel relevant and increase their feelings of personal fulfillment, thus avoiding burnout risk situations. Our results indicate that this aspect is more relevant for the worker than the rewards that he/she receives for his/her work. It is also important to ensure job stability with long-term contracts. It is also important to highlight that companies should value the most scrupulous workers because they are the ones who will show the most commitment to their job and will therefore be more motivated to do a good job. Finally, it is necessary to highlight the value that older employees bring to the organization, avoiding situations of discrimination based on age or ageism.

**Author Contributions:** S.R.-V.: Conceptualization, Methodology, Formal Analysis, Resources, Investigation, Data Management, Writing, Project Administration, Visualization; E.M.D.-R.: Conceptualization, Methodology, Formal Analysis, Investigation, Writing, Supervision, Visualization; M.I.L.-N.: Supervision, Visualization. All authors have read and agreed to the published version of the manuscript.

**Funding:** This research received no external funding.

**Institutional Review Board Statement:** The study was conducted in accordance with the Declaration of Helsinki and approved by the Institutional Ethics Committee of Faculty of Psychology of Universidad Complutense de Madrid (Ref.: 2019/20-022, 2019) for studies involving humans.

**Informed Consent Statement:** Informed consent was obtained from all subjects involved in the study.

**Data Availability Statement:** The data presented in this study are available on request from the corresponding author.

**Conflicts of Interest:** The authors declare no conflict of interest.

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
