# Peer review of "What Does the Feeling of Job Success Depend On? Influence of Personal and Organizational Factors"

_societies, doi:10.3390/soc13060140_

Round 1

Reviewer 1 Report

The article tackles an interesting topic. Additionally the results which are presented in the paper are valuable for better understanding of the phenomenon. The only comment, in Table 2 to add the Significance level, but also to highlight the important correletations.

Author Response

Reviewer 1.

The article tackles an interesting topic. Additionally the results which are presented in the paper are valuable for better understanding of the phenomenon. The only comment, in Table 2 to add the Significance level, but also to highlight the important correletations.

RE: Thank you very much for your review and comments. Following your indications, we have included the statistical significance in Table 2 and we have highlighted (in bold) the highest values of the correlations with success.

Reviewer 2 Report

General comments

While the abstract provides a good overview of the paper's main arguments and findings, it could benefit from more specific details about the methodological approach, novelty, and recommendations of the study. The authors recommended including methodological approach, novelty, and recommendations of the study

Introduction

I would say, the introduction part is good but scanty. The introduction part is lacking to provide a clear and concise overview or background information about what has been studied before and why this topic is important to investigate. It should show us how the Influence of personal and organizational factors affects job success. The authors should include some pieces of literature conducted on this area with their finding  

The authors should include section 2 as “Related Literature Review”

Methodology and Materials should be in section 3

The methodology section lacks a research approach and design, target population, sampling techniques, and sample size

Results

The discussion is more of a presentation of results than a discussion. As a result, an attempt should be made to explain the findings in light of the literature presented, as well as the intrinsic value of the findings for the research scenario.

The authors should combine the result and discussion section to provide a clear understanding

The finding and conclusions must be supported by previously established studies

Author Response

Reviewer 2.

General comments

While the abstract provides a good overview of the paper's main arguments and findings, it could benefit from more specific details about the methodological approach, novelty, and recommendations of the study. The authors recommended including methodological approach, novelty, and recommendations of the study.

 RE: We have modified the abstract, as suggested by the reviewer

Introduction

I would say, the introduction part is good but scanty. The introduction part is lacking to provide a clear and concise overview or background information about what has been studied before and why this topic is important to investigate. It should show us how the Influence of personal and organizational factors affects job success. The authors should include some pieces of literature conducted on this area with their finding.

RE: We have expanded the introduction, as suggested by the reviewer

The authors should include section 2 as “Related Literature Review” Methodology and Materials should be in section 3.

RE: When preparing our manuscript, we have followed the indications of the template provided by the journal Societies (Microsoft Word: Manuscripts prepared in Microsoft Word must be converted into a single file before submission. When preparing manuscripts in Microsoft Word, we encourage you to use the Societies Microsoft Word template file. Please insert your graphics (schemes, figures, etc.) in the main text after the paragraph of its first citation. All text and metadata identifying the authors should be removed before submission.), which indicates that the sections are: 1. Introduction 2, Materials and methods, 3. Results, 4. Discussion, 5. Conclusions, 6. Patents. However, following this comment from the reviewer, we have renumbered the sections as directed and added a new section 2.Related literature review.

The methodology section lacks a research approach and design, target population, sampling techniques, and sample size.

RE: We have added new information on these methodological issues

The discussion is more of a presentation of results than a discussion. As a result, an attempt should be made to explain the findings in light of the literature presented, as well as the intrinsic value of the findings for the research scenario. The authors should combine the result and discussion section to provide a clear understanding. The finding and conclusions must be supported by previously established studies.

 RE: We have expanded the discussion and conclusions, as suggested by the reviewer

Round 2

Reviewer 2 Report

Congratulations